# Quantum coherent manipulation and readout of superconducting vortex states

Ameya Nambisan[1], Simon Günzler[1,2], Dennis Rieger[1,2], Nicolas Gosling[1], Simon Geisert[1], Victor Carpentier[1], Nicolas Zapata[1], Mitchell Field[1], Milorad V. Milošević[3], Carlos A. Diaz Lopez[4], Ciprian Padurariu[4], Björn Kubala[4,5], Joachim Ankerhold[4], Wolfgang Wernsdorfer[1,2], Martin Spiecker[1,2] & Ioan M. Pop[1,2,6 ✉]

A defining characteristic of superconductors is their tendency to expel magnetic fields, yet above a critical threshold, magnetic flux penetrates in discrete quanta carried by Abrikosov vortices[1]. The superconducting gap is completely suppressed at the vortex core, rendering them dissipative, semi-classical entities that impact applications from high-current-density wires to quantum devices. Material disorder can drive a crossover to vortices that preserve an energy gap at the core[2–4], owing to intrinsic[5] or emergent granularity on the scale of the coherence length[2,6]. Although quantum vortex behaviour could emerge in this effective tunnel-junction regime[7], and signatures have been observed in diverse systems[8–10], coherent manipulation of vortex states has remained elusive. Here we present evidence that vortices trapped in granular superconducting films can behave as two-level systems, exhibiting microsecond-range quantum coherence and energy relaxation times that reach fractions of a millisecond. Using the tools of circuit quantum electrodynamics[11], we perform coherent manipulation and quantum non-demolition readout of vortex states in granular aluminium microwave resonators, heralding future directions for quantum information processing, materials characterization and sensing.

From the moment of its discovery, the antagonistic relation between superconductivity and a magnetic field has provided a complex playground for experimentalists and theorists alike. The measurement of the critical field and the Meissner effect[12] have anchored phase-transition theories[13,14], and the trapping of quantized flux inside superconductors has provided direct evidence for the existence of Cooper pairs[15]. A hallmark of type II superconductivity in a magnetic field is the formation of Abrikosov vortices: regions of local gap suppression that interact to form lattices[1]. Vortex dynamics is detrimental for a wide range of applications[15], causing heating, flux noise and magnetic hysteresis. However, pinned vortices enable quasiparticle trapping in their core, which enhances the critical current[16] of superconducting films, improves micro-cooler efficiency[17], boosts resonator quality factors[18] and improves qubit coherence[19,20]. In all these cases, owing to the normal state core, vortices can be understood within semi-classical models.

Gap suppression in the vortex core stems from the crowding of supercurrent at its centre, a consequence of continuity in the superconducting medium. Recent work[3] has proposed that in discretized systems, such as granular superconductors where non-superconducting regions separate superconducting islands, the vortex core can remain gapped and dissipationless; a closely related regime has also been predicted for strongly disordered superconductors, where emergent superconducting islands[2] host vortices with insulating cores[4]. Although quantum behaviour has been revealed by tunnelling of vortices in long Josephson junctions[8] and thin films[9], or via the zero-point motion of pinned vortices[10], direct evidence of coherent superconducting vortex states has yet to be observed.

Here we show that vortices trapped in a superconducting granular aluminium (grAl) microwave resonator form field-tunable two-level systems that behave like effective spins, strongly coupled to the resonator. They can therefore be regarded as quantum bits (qubits) that arise from vortex tunnelling in a field-modulated double-well potential formed between pinning sites. These vortex qubit (VQ) states exhibit microsecond coherence and energy relaxation times on the order of $10^2$ μs, strikingly different from the dissipative dynamics of Abrikosov vortices. We find that VQs remain stable for weeks, enabling coherent control and quantum non-demolition readout within the framework of circuit quantum electrodynamics[11].

As schematized in Fig. 1, we use a grAl micro-stripline resonator, with resistivity $\rho = 3,600$ μΩ cm, chosen to be within a factor of 3 below the superconducting-to-insulating transition[21]. In this regime, the film consists of Al grains of 3–4-nm diameter separated by amorphous $AlO_x$ barriers, resulting in a coherence length $\xi \approx 7$ nm and London penetration depth of $\lambda_L \approx 4$ μm (refs. 5,22,23). The resonator is placed in a cylindrical copper waveguide (Supplementary Information section I)

[1]IQMT, Karlsruhe Institute of Technology, Karlsruhe, Germany. [2]PHI, Karlsruhe Institute of Technology, Karlsruhe, Germany. [3]COMMIT, Department of Physics, University of Antwerp, Antwerp, Belgium. [4]Institute for Complex Quantum Systems and IQST, University of Ulm, Ulm, Germany. [5]German Aerospace Center (DLR), Institute of Quantum Technologies, Ulm, Germany. [6]Physics Institute 1, Stuttgart University, Stuttgart, Germany. ✉e-mail: ioan.pop@kit.edu

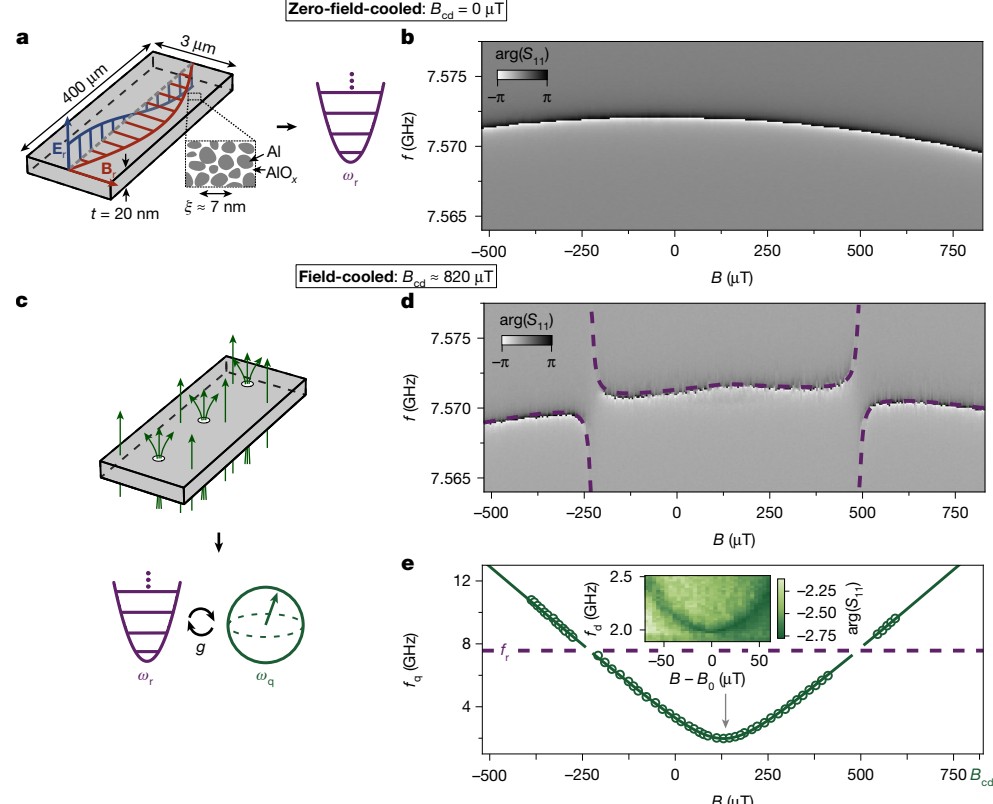

**Fig. 1 | Field cooling introduces VQ states that couple to the grAl resonator.**
**a**, When cooled to 20 mK in perpendicular magnetic field $B_{cd} = 0$ μT, a $\lambda/2$ micro-stripline grAl resonator behaves as a quantum harmonic oscillator with resonant frequency $\omega_r$. The electric- and magnetic-field distributions are illustrated in blue and red, respectively. The grAl film has a thickness of $t = 20$ nm and a superconducting coherence length of $\xi = 7$ nm. **b**, Phase response $\arg(S_{11})$ of the resonator measured in reflection, as a function of perpendicular magnetic field $B$ applied after cooling. The measured parabolic suppression of the resonance is given by the increase in kinetic inductance owing to screening currents[25], and the field range is limited by the vortex penetration threshold[26]. **c**, When cooled in perpendicular magnetic field $B_{cd} = 820$ μT (see main text), vortices enter the grAl resonator and the system exhibits a behaviour akin to a

flux qubit with a transition frequency $\omega_q$ coupled to a readout resonator, as illustrated in **d** and **e**. **d**, The measured phase response of the resonator as a function of $B$ reveals avoided level crossings, suggesting coupling to vortex states. The purple dashed line shows a fit to the asymmetric quantum Rabi model (equation (2)), yielding the coupling $g/2\pi = 95$ MHz. **e**, Extracted VQ frequency $f_q$ from two-tone spectroscopy (see inset) as a function of $B$. The green line corresponds to the joint fit of data in **d** and **e** to equation (2), and the purple dashed line marks the bare resonator frequency $f_r$. Inset: two-tone spectroscopy in the vicinity of $B_0$ corresponding to the minimum frequency of the VQ. The colour scale indicates the measured phase response as a function of the frequency $f_d$ of the second drive.

anchored to the 20-mK base plate of a dilution cryostat and measured in reflection. When cooled in zero magnetic field $B_{cd} = 0$ μT, the grAl resonator behaves as a weakly anharmonic oscillator[24], with a fundamental frequency $f_r = 7.572$ GHz, set by its dimensions (3 μm wide, 400 μm long; Extended Data Fig. 1). Figure 1b shows the frequency decrease with perpendicular magnetic field $B$, as expected with the increase in kinetic inductance[25,26].

Following field-cooling, sweeping $B$ reveals avoided level crossings in the grAl resonator response as illustrated in Fig. 1d, which we interpret as evidence of strong coupling with $g/2\pi = 95$ MHz to vortex states. To extract the mode's spectrum, we sweep a second microwave drive while probing the readout resonator (Fig. 1e). We observe a minimum vortex mode frequency $f_q = 2$ GHz at the sweet spot $B_0 = 128$ μT (Fig. 1e, inset), with a slope of the hyperbolic field dispersion $\gamma = 20$ GHz mT$^{-1}$, reminiscent of a flux qubit[27]. As the field approaches the sweet spot, the resonance narrows, pointing to magnetic-field fluctuations as dominant noise source[28]. From measured spectra across 32 field-cooling cycles in six different resonators, we extract values of $g, f_q, B_0$ and $\gamma$ that are of similar order of magnitude but vary between cycles (Supplementary Information section II), suggesting different underlying vortex configurations. Repeated resonator reflection coefficient $S_{11}$ measurements at the sweet spot reveal two distinct clusters in the quadrature plane (Fig. 2a), indicating that the vortex state has a lifetime well beyond

the 1.2-μs integration time, thereby enabling single-shot state discrimination. As demonstrated in Fig. 2b, by driving at $f_q$, we can calibrate a 20 ns π-pulse, which inverts its thermal population (see Supplementary Information section III for the Rabi oscillations). These signatures define the VQ states $|g\rangle$ (ground) and $|e\rangle$ (excited). From their steady-state populations, we extract a 74-mK effective temperature. The VQ–resonator interaction induces a state-dependent dispersive shift $\chi/2\pi = f_{r,|e\rangle} - f_{r,|g\rangle}$. As shown in Fig. 2c, fitting the resonator's phase response to the centres of in-phase and quadrature (IQ) clouds measured versus readout frequency yields $\chi/2\pi = -1.32$ MHz (see Supplementary Information section IV for all measured IQ clouds).

For further insight into the nature of the VQ and its interaction with the grAl resonator, we measure $\chi$ versus field, as shown in Fig. 2d. We model it using the quantum Rabi model (QRM) for a spin $S = 1/2$ coupled via $\mathcal{H}_c = \hbar g(\hat{a}^\dagger + \hat{a})\sigma_x$ to a harmonic oscillator with frequency $\omega_r$ and Hamiltonian $\mathcal{H}_r = \hbar\omega_r\left(\hat{a}^\dagger\hat{a} + \frac{1}{2}\right)$ (Supplementary Information section V). Here $\hat{a}^\dagger$ and $\hat{a}$ are the resonator bosonic operators, $\hbar = h/(2\pi)$ is the reduced Planck constant and $\sigma_x$ is the Pauli matrix for a spin $\mathbf{S} = \hbar/2\boldsymbol{\sigma}$. The interaction energy between the spin and the magnetic field is $\gamma\mathbf{S}\cdot(\widetilde{\mathbf{B}} + \mathbf{B}')$, where $\gamma$ is the gyromagnetic ratio and the field consists of two contributions: a pseudo-field $\widetilde{\mathbf{B}}$ that sets the VQ energy at the sweet spot, and the applied magnetic field $|\mathbf{B}'| = B - B_0$ measured from the sweet spot. We compare joint fits of the measured VQ and

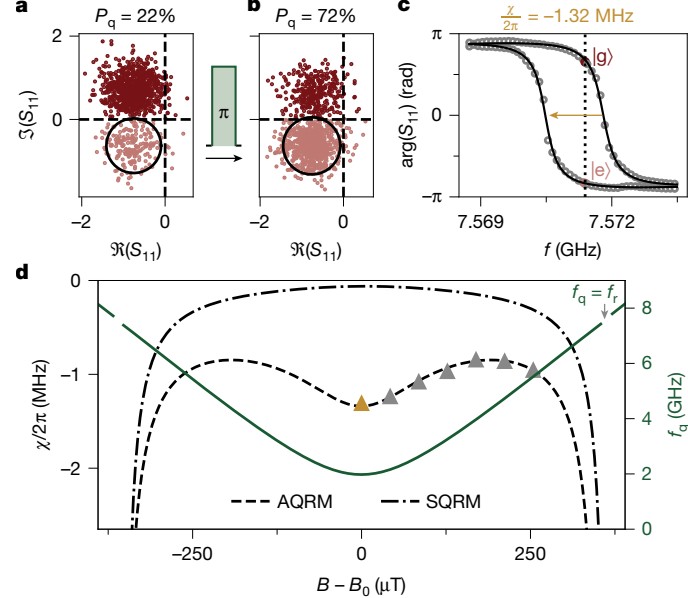

resonator frequencies in field (Fig. 1d,e), using the symmetric quantum Rabi model (SQRM)

$$\mathcal{H}_{\mathrm{SQRM}} = \mathcal{H}_{\mathrm{r}} + \mathcal{H}_{\mathrm{c}} + \frac{\hbar\gamma}{2}\sigma_z\sqrt{\widetilde{B}^2 + B'^2}, \tag{1}$$

and the asymmetric quantum Rabi model (AQRM)

$$\mathcal{H}_{\mathrm{AQRM}} = \mathcal{H}_{\mathrm{r}} + \mathcal{H}_{\mathrm{c}} + \frac{\hbar\gamma}{2}\sigma_z\widetilde{B} - \frac{\hbar\gamma}{2}\sigma_x B'. \tag{2}$$

Only the AQRM captures the non-monotonic dependence of $\chi$ with $B$. In contrast, the SQRM predicts a monotonically decreasing $\chi$ with detuning from the resonator. Moreover, using the coupling constant $g$ from the joint fit in Fig. 1d,e, we obtain quantitative agreement for the measured $\chi$, as shown in Fig. 2d. This suggests that the VQ, possibly consisting of persistent currents, arises from dynamics in a double-well potential, analogous to fluxon tunnelling through the Josephson junction of a flux qubit[27]. Within this model, the pseudo-field $\widetilde{B}$ is given by the fluxon tunnelling amplitude[7].

We complete the characterization of the VQ with time-domain measurements at the sweet spot. As shown in Fig. 3a, the fitted energy relaxation time is $T_1 = 186\ \mu s$, with values ranging from 40 μs to 300 μs across multiple VQ preparation cycles (Supplementary Information section VI). Relaxation times extracted from VQ quantum jumps (Supplementary Information section VI) fall within the temporal fluctuations observed in free decay, indicating a quantum non-demolition readout. Remarkably, the VQ exhibits quantum coherence, with a Ramsey time $T_2^* = 440\ ns$, which extends to $T_2^{\mathrm{echo}} = 1.2\ \mu s$ in Hahn-echo measurements, which suppress the low-frequency noise (Fig. 3b,c). The Ramsey fringes exhibit a beating pattern, corresponding to a toggling of the VQ's frequency between two values separated by 1.9 MHz. This feature is sometimes also observed in superconducting qubits[29], possibly indicative of charge noise or conductance channel fluctuations. The measured VQ lifetime $T_1$ is competitive with superconducting flux qubits[30,31], whereas the coherence $T_2^*$, $T_2^{\mathrm{echo}}$ remains more modest, in line with flux qubit devices realized entirely from disordered superconductors[29,32]. Away from the sweet spot, both $T_2^*$ and $T_2^{\mathrm{echo}}$ decrease (Supplementary Information section VI), consistent with flux-noise-limited dephasing in loop-based superconducting circuits and motivating a detailed comparison with established flux-noise

**Fig. 2 | The asymmetric quantum Rabi model describes the VQ dispersively coupled to its resonator. a**, Consecutive $S_{11}$ measurements at the sweet spot show two IQ clouds in the complex plane. The relative occurrence of points in the clouds corresponds to the population of the $|g\rangle$ (ground) and $|e\rangle$ (excited) states. The qubit excited state population $P_q$ yields an effective qubit temperature $T_{\mathrm{eff}} \approx 74\ mK$. **b**, Measured IQ clouds following a 20-ns drive at $f_q$ calibrated to implement a π-pulse show a population inversion as expected for a two-level system. The black circles have a radius of 1.5 standard deviation. **c**, Resonator phase response $\arg(S_{11})$, obtained from the centres of the IQ clouds, measured versus readout frequency $f_{\mathrm{RO}}$ in the vicinity of $f_r$. A fit to the data (black solid line) yields a dispersive shift of $\chi/2\pi = -1.32\ MHz$. The dark red ($|g\rangle$) and light red ($|e\rangle$) points correspond to the data in **a** at $f_{\mathrm{RO}} = 7.5714\ GHz$ (dashed line). **d**, Variation of $\chi$ with magnetic field $B$, shown as triangles, with the yellow triangle corresponding to the measurement in **b**. The dashed line indicates the expected values from the asymmetric quantum Rabi model equation (2) with $g_{\mathrm{AQRM}}/2\pi = 92.5\ MHz$, and the dash-dotted line to the symmetric quantum Rabi model equation (1) with $g_{\mathrm{SQRM}}/2\pi = 20\ MHz$. The solid green line represents the qubit frequency (right axis), similar to Fig. 1d.

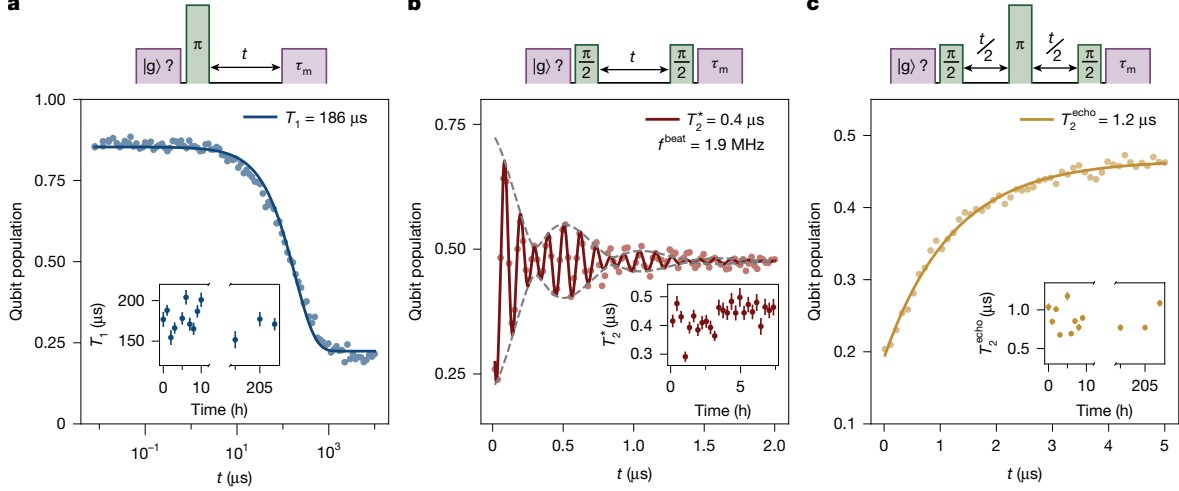

**Fig. 3 | Measurement of low loss and coherence in the VQ. a**, Free energy decay measured after a 20-ns π-pulse applied selectively to the VQ measured in the ground state $|g\rangle$. The readout pulse has a duration $\tau_{\mathrm{m}} = 1.2\ \mu s$. The excited VQ population as a function of wait time $t$ is fitted with an exponential corresponding to $T_1 = 186\ \mu s$ (solid line). **b**, Ramsey fringes exhibit a beating pattern, resulting from two frequencies separated by $f_{\mathrm{beat}} = 1.9\ MHz$. We extract $T_2^*$ Ramsey coherence times of 440 ns. **c**, Spin Hahn-echo measurement with extracted $T_2^{\mathrm{echo}} = 1.2\ \mu s$. For each panel, the corresponding pulse sequence is sketched at the top, and the insets show measured coherence times, with error bars indicating the standard deviation from the fit, over several hours.

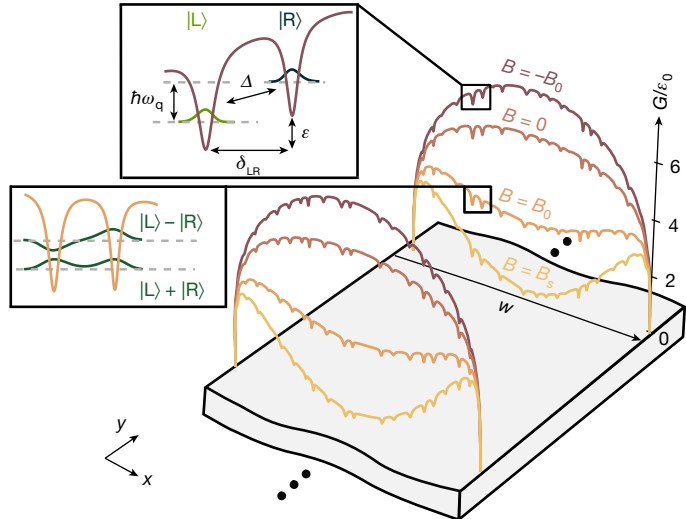

**Fig. 4 | Origin of the double-well potential.** Gibbs free energy $G_1$ (equation (3), baseline) of a single vortex, shown with added pinning potentials modelled as Lorentzian dips, in units of $\varepsilon_0 = \Phi_0^2/2\pi\mu_0\Lambda \approx 2$ THz. The vortex position is measured from the edge, as indicated by the coordinate axis. Colours represent different applied magnetic fields from $B_s = \phi_S\Phi_0/w^2$ to $-B_0$. Top inset: example of a double-well potential formed by the energy landscape of adjacent pinning sites separated by $\delta_{LR}$ and offset in energy by $\epsilon$. The localized wavefunctions correspond to the two vortex positions $|L\rangle$ and $|R\rangle$, coupled by tunnelling amplitude $\Delta$, with an energy splitting of $\hbar\omega_q$. Bottom inset: at the sweet spot ($B_0$, which can be higher or lower than $B_s$; see Supplementary Information section II), the double well is degenerate, with VQ states forming symmetric and antisymmetric combinations of the localized wavefunctions, yielding $\hbar\omega_q = 2\Delta$.

mechanisms[30,31,33]. In future experiments, detailed noise characterization[31], environment polarizability[34], as well as susceptibility to in-plane magnetic[33] and electric fields[35] could shed light on the microscopic origin of the VQ and its environment.

To give a hypothesis for the origin of the double-well potential of the VQ, we consider the process of introducing vortices into the grAl resonator through field-cooling. Their formation and spatial arrangement depend on the value of the flux bias during cooling $\phi = B_{cd}w^2/\Phi_0$, where $\Phi_0 = h/2e$ is the magnetic flux quantum, $e$ is the charge of an electron, and $w$ is the width of the resonator. In the Pearl limit[36], where the thickness of the film $t \ll \lambda_L$, the threshold for stable vortices is $\phi_S = (2/\pi)\ln(2w/\pi\xi)$ (refs. 37–39), corresponding to $\phi_S = 3.59$ for our geometry (Supplementary Information section VII). The Gibbs energy for vortices threading the film[40,41] is

$$G_1(x) = \varepsilon_0\ln\left(\frac{2w}{\pi\xi}\sin\left(\frac{\pi x}{w}\right)+1\right) - \frac{\Phi_0(B-n\Phi_0)}{\mu_0\Lambda}x(w-x), \tag{3}$$

where $\varepsilon_0 = \Phi_0^2/(2\pi\mu_0\Lambda)$ sets the single-vortex energy scale, $n$ is the density of vortices ($n = 0$ for the first vortex), $\Lambda = 2\lambda_L^2/t$ is the Pearl length of the resonator, and $x$ is the position of the vortex measured from the resonator edge. As $B$ decreases from $B_s = \phi_S\Phi_0/w^2$ to zero, the minimum of $G_1(x)$ vanishes (Fig. 4, baseline), and in the absence of pinning the vortex would be expelled.

To account for the measured stability of the VQ across magnetic-field sweeps (Fig. 1), we incorporate pinning potentials, presumably abundant given the disordered nature of grAl. They are modelled by adding Lorentzian dips $V_{pin} = V_i(1 + (x-x_i)^2/\sigma_i^2)^{-1}$ to $G_1(x)$, at random positions $x_i$, depth $V_i$ and width $\sigma_i$, sketched as the coloured energy landscapes in Fig. 4. A vortex tunnelling between pinning sites forms a double-well potential (Fig. 4, top inset), in which $B$ tunes the relative pinning depths according to equation (3). At $B_0$, the minima are degenerate

and the vortex delocalizes, with $|g\rangle$ and $|e\rangle$ given by symmetric and antisymmetric superpositions of $|L\rangle$ and $|R\rangle$ wavefunctions (Fig. 4, bottom inset).

This hypothesis is supported by the fact that typically measured gyromagnetic ratios $\gamma/2\pi = 3$–25 GHz mT$^{-1}$ are consistent with flux tunnelling between pinning sites separated by tens of nanometres (Supplementary Information section VII), reminiscent of tunnelling through grAl nanojunctions[29]. Moreover, to leading order, a kinetic-inductance-mediated VQ–resonator coupling $g/\omega_r \approx 0.1$–1% (Supplementary Information section VIII) is consistent with the observed avoided level crossings. Although single-vortex pinning can account for the observed VQ, it is well established that multiple vortices simultaneously enter the resonator once the threshold for entry is reached[39], as illustrated by the set of Gibbs curves in the foreground of Fig. 4. We estimate the VQ–VQ interaction in the 10–100 MHz range (Supplementary Information section IX), suggesting that collective vortex dynamics is unlikely. Nevertheless, distinguishing between single- and multi-vortex dynamics, for instance, using imaging methods[40,42–45], or by shaping the resonator width[18], remains an important avenue for future research.

In conclusion, field-cooling a grAl micro-stripline resonator reproducibly generates VQ states that couple dispersively to the resonator and can be coherently driven. Our results demonstrate that superconducting vortices can harbour quantum coherence on microsecond timescales. Remarkably, the VQ energy relaxation times are on the order of hundreds of microseconds, comparable to those of engineered superconducting qubits[11,30], and qualitatively distinct from the dissipation expected for Abrikosov vortex dynamics. This supports a picture of grAl as a three-dimensional network of Josephson junctions, expected to host gapful-core vortices once the coherence length $\xi$ becomes comparable to the intergrain spacing $\ell$, with a growing minigap for $\xi \lesssim \ell$ (refs. 2,3). The observed dispersive shifts and spectra are accurately captured by an asymmetric quantum Rabi model, consistent with a two-level system in a double-well potential. Microscopically, this may arise from vortex tunnelling between pinning sites, modulated by the magnetic-field dependence of the Gibbs energy. This hypothesis, although consistent with our measurements, remains to be confirmed by future experiments such as scanning tunnelling or scanning superconducting quantum interference device (SQUID) microscopy.

Looking ahead, the measurement of quantum coherence in vortex states, along with their relative technological simplicity, opens several exciting avenues in quantum science. Disordered superconductors beyond grAl[46,47] or engineered two-dimensional networks of Josephson junctions[48] may host similar VQs, shedding light onto the complex physics in the vicinity of the superconductor-to-insulator transition[49,50]. Moreover, this would offer an embedded tool for material characterization at the microscopic level. In the same spirit, if the observed dynamics indeed stem from single-vortex tunnelling, VQs could be harnessed for nanoscale sensing. Ultimately, engineering the pinning landscape and device geometry, combined with noise spectroscopy and susceptibility measurements to magnetic and electric fields, will be crucial to enhance VQ coherence and possibly launch a vortex-based quantum information platform.

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

## Methods

We fabricated the sample on a double-side polished $c$-plane sapphire substrate using electron-beam lithography and wet etching. The substrate was cleaned in a 50 °C acetone bath and rinsed in ethanol, before applying an $Ar/O_2$ ion descum process using a Kaufman ion source in a PreVac evaporation system. After performing titanium gettering, we deposited a 20-nm grAl film at room temperature by evaporating aluminium at 1 nm s$^{-1}$ under dynamic oxidation. The resulting film had a sheet resistance of 1.5 k$\Omega$ $\square^{-1}$. To define the resonator geometry, we patterned a 300-nm-thick resist layer (ARN 7520.18) using a 50-keV e-beam writer. We developed the resist for 40 s in an AR 300-47:$H_2O$ mixture (4:1), followed by wet etching in MF 319. Extended Data Fig. 1 shows microscopy of the resulting grAl resonator structures. The resonator had a uniform height of 24 nm with slanted edges.

### Data availability

The data supporting this study are available via Zenodo at https://doi.org/10.5281/zenodo.18207758 (ref. 51).

### Code availability

The analysis scripts used in this study are are available via Zenodo at https://doi.org/10.5281/zenodo.18207758 (ref. 51).

51.   Nambisan, A. et al. Data and code for the article "Quantum coherence in superconducting vortex states". *Zenodo* https://doi.org/10.5281/zenodo.18207758 (2026).

**Acknowledgements** We thank M. Feigel'man, L. Glazman, G. Kirchmair, N. Roch, A. Shnirman, U. Vool and W. Wulfhekel for discussions; and we acknowledge the technical support from L. Radtke and S. Diewald. We acknowledge funding from the Baden-Württemberg Stiftung under the QT-10 project (QEDHiNet). The KIT Nanostructure Service Laboratory provided support for the facilities used. We recognize the qKit measurement software framework. We acknowledge support by the state of Baden-Württemberg through the bwHPC. M.S. and N.G. acknowledge support from the Federal Ministry of Research, Technology, and Space (BMFTR) within the GEQCOS project (FKZ: 13N15683). N.Z. acknowledges funding from the Deutsche Forschungsgemeinschaft (DFG - German Research Foundation) under project number 450396347 (GeHoldeQED). S. Günzler, D.R. and W.W. acknowledge support from the Leibniz award WE 4458-5. C.P. and J.A. acknowledge support from the Carl-Zeiss-Stiftung via the QPhoton centre as well as through the Deutsche Forschungsgemeinschaft (DFG) via AN336/13-1. M.V.M. acknowledges support from the Research Foundation-Flanders (FWO).

**Author contributions** A.N., S. Günzler, D.R., M.S. and I.M.P. conceived of and designed the experiment. A.N. and V.C. contributed to the device fabrication. A.N., S. Günzler, D.R., N.G., S. Geisert, V.C., N.Z., M.F. and M.S. participated in the measurements. A.N., S. Günzler., D.R., N.G. and M.S. analysed the data. A.N., S. Günzler and I.M.P. led the paper writing, and all authors contributed to the text. A.N., M.V.M., C.A.D.L., C.P., B.K., M.S. and I.M.P contributed to theoretical modelling. J.A., W.W., M.S. and I.M.P. supervised the project.

**Funding** Open access funding provided by Karlsruher Institut für Technologie (KIT).

**Competing interests** The authors declare no competing interests.

**Additional information**
**Correspondence and requests for materials** should be addressed to Ioan M. Pop.

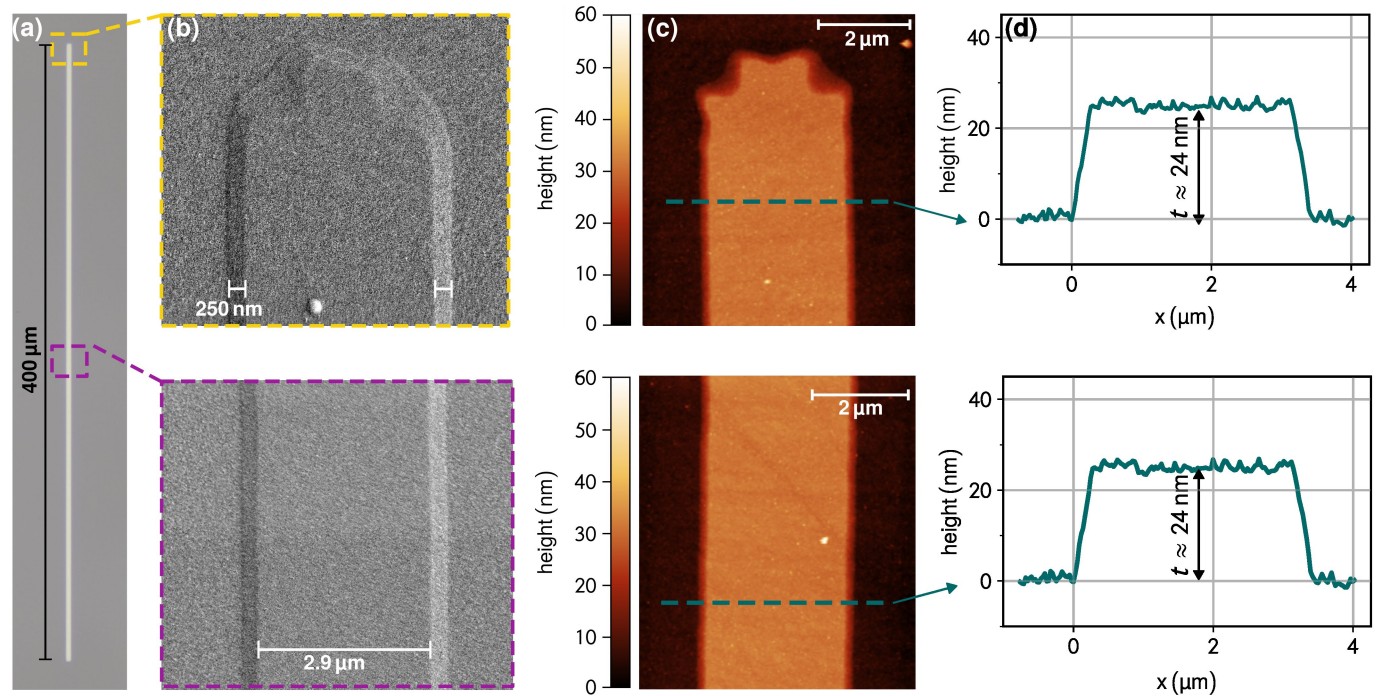

**Extended Data Fig. 1 | Microscopy of the grAl resonator. (a)** Optical micrograph showing the 400 µm long resonator. **(b)** Scanning electron microscope (SEM) image of the resonator's end (top) and center (bottom) reveals 250 nm wide slanted edges. **(c)** Atomic force microscopy (AFM) image acquired in tapping mode (tip radius < 8 nm). **(d)** The AFM line profile yields a film thickness of $t = 24$ nm.