## [Peer Review File · Nature]

Quantum coherent manipulation and readout of superconducting vortex states

Corresponding Author: Ms Ameya Nambisan

Version 0:

Reviewer comments:

Referee #1

(Remarks to the Author)

The paper contains important results that are especially important for the growing field of quantum coherence and manipulation. However, the paper is not well written for reasons I outline below.

The authors would like to claim that the vortices in granular aluminum (grAl) behave like qubits, and not the standard Abrikosov vortices with dissipative metallic cores. This feature results in long coherence times and therefore enables various functionalities.

[1] To get this message across it is necessary to first describe the experimental results without any reference to theory.

[2] It will be useful to have a comparison of various time scales (coherence, switching) in current platforms, a table will help. This will help position grAl as a new platform for certain functionalities.

[3] The main results of the paper should be described at the beginning before the experimental system and measurements are described. These should also be highlighted in the abstract.

[4] The role of theory is best described after the experiments. It is important to note that the predictions of vortices with insulating cores have been made by other groups as well in disordered superconductors and these should be cited. In these cases disorder creates emergent granularity that behaves very similarly to grAl.

[5] It will be useful to compare with disordered superconductors and the coherence time scales of Josephson vortices trapped with insulating cores.

Referee #2

(Remarks to the Author)

The manuscript of Nambisan et al. describes intriguing measurements of granular Al (grAl) microwave resonators in magnetic fields. For sufficiently large magnetic fields, the authors observe a microwave response of the resonator that behaves in a quite similar way to a Josephson-junction based flux qubit coupled to a resonator. This includes coherent Rabi oscillations characteristic of a driven two-level system, as well as measurements of energy relaxation times, which are on par with good conventional superconducting qubits, and dephasing rates. However, the device measured here is elegantly simple, merely a narrow strip of grAl with no microfabricated Josephson junctions. The results are interpreted in terms of a model of vortices trapped in random pinning wells, with one vortex able to tunnel between neighboring wells while coupling to the zero-point currents in the microwave resonator. Unlike Abrikosov vortices in conventional superconductors, where dissipation in the vortex core disfavors such quantum coherent effects, a recent theory for vortices in grAl films, described in Ref. [1], predicts a mini-gap in the vortex core bound states and thus greatly reduced dissipation.

The results presented here are rather amazing. The possibility of quantum mechanical effects in the motion of a vortex in a superconductor have been debated for many years, with some prior evidence for incoherent vortex tunneling but nothing demonstrating any sort of coherence that I am aware of. The clear two-level behavior measured here demonstrates that something quite surprising is happening in the vortex state of this grAl resonator. Although the authors present a mostly plausible model for what is happening with the vortices to explain their measurements, I am not completely convinced by all

of the details of the model. Nonetheless, the results are so striking, and, for the most part, carefully executed, and I believe will inspire further experimental work and significant theoretical investigations, potentially addressing long outstanding questions such as the effective mass of a vortex in a superconductor. Thus, I recommend publication following a series of edits that I describe below.

*My primary concern is with the interpretation in terms of the tunneling dynamics of a single vortex. The authors argue in Appendix I that intervortex interactions are relatively weak so that the motion of the vortices can be considered independently. Each time the resonator is cycled above T_c and cooled in magnetic field B_{cd} , there are of order 100 vortices trapped in the resonator. The picture presented is that most of the time just one of these ~ 100 vortices exhibits quantum coherent tunneling dynamics that couples to the resonator. With each repeated magnetic field cycling above T_c in the same cooling field B_{cd} , the authors observe different parameters, such as tunneling energy, field dispersion, and sweet spot offset for the VQ (vortex qubit). The fact that there are different values for each field cycle seems reasonable since the vortex trapping locations will depend on microscopic details of the pinning configuration in these highly disordered films as well as details of the exact thermal trajectory for the field cycle, which I suspect is difficult to reproduce exactly. Thus, many of the ~ 100 vortices will likely be trapped either in wells that are too deep or too far away from adjacent wells to allow for any tunneling. What seems surprising is that the authors report primarily seeing one prominent VQ for each field cycle, with only occasional field cycles resulting in zero or multiple VQs. In this independent vortex picture, I would have expected many cycles with no VQs and some with multiple VQs, with single-VQ cycles being not especially favored. Can the authors be more quantitative? If there are enough field cycles of the same device, would it be possible to make a histogram of the number of VQs observed on each cycle? It would also be useful to have in the Appendix some plots equivalent to Fig. 1(d) for an example cycle with 0 VQs and another with 2 (or more) VQs.

*The value of $B_{cd} = 820$ uT is indeed what one would calculate from the standard expression for the threshold cooling field from Refs. [30,31] for $\xi = 7$ nm, but was there an attempt to check this experimentally, such as repeated scans like Fig. 1(b,d) for different cooling fields? I appreciate that cycling above and below T_c in different cooling fields is time consuming and nontrivial on a dilution fridge, but it would be interesting to see if there are at least some cooling fields below 820 uT that don't result in the trapping of any vortices. Also, I would expect that there should be a downwards frequency shift in the resonance when the cooling field is greater than the threshold field, even if there are no VQs, coming from the reactance of the trapped vortices.

*I don't fully understand the authors' statement on line 100. It seems from their earlier arguments that cooling in $\phi = \phi_s$ should just result in a number of vortices per square of 1, not a range from 1-3. This should be clarified.

*In Fig.1 and related scans, do the authors ever observe hysteresis if the B-field sweep is retraced, or are the ranges of these scans always chosen to stay below the entry/exit field for vortices to jump in/out of the resonator? This is briefly mentioned in the caption, but it would be good to add detail if possible. The authors describe complementary ZFC measurements where they can inject vortices into an initially flux-free state, which makes sense. But it should also be possible to change the vortex density for an FC state by varying the B field over a sufficiently wide range.

*I don't completely follow the authors' derivation for the VQ-resonator interaction in Appendix H. It appears that this indicates that the coupling will be maximized for the situation with the two pinning sites of the VQ oriented along the y direction, i.e., the length of the resonator, and zero for an orientation along x. My intuition would be that the tunneling between pinning sites along the resonator width direction, i.e., x direction, would give the strongest g, since zero point currents in the resonator would exert a Lorentz force on the vortex that would tilt the bias of one pinning well relative to the other. This is also how the authors have drawn the example pinning sites in Fig. 2(e). I realize there are many unknown quantities in the derivation in Appendix H and this is mainly intended to provide a plausible argument for the order of magnitude of g, but it would be good to clarify this point about the orientation of the pinning sites for a VQ.

*In the appendix, it would be helpful to have a schematic of the dilution fridge setup and wiring. Was any magnetic shielding included to screen external magnetic fields so that zero current in the Helmholtz pair is close to zero total field? Was any IR shielding used around the waveguide containing the grAl resonator? This seems potentially challenging with the Helmholtz pair at 4K, but perhaps could be related to the elevated effective temperatures for the VQs.

*In Eq. (3), and also Eq. (G2), where does the +1 in the argument of the log term in Eq. (3) come from? This is not in Ref. [31], although it doesn't make much of a difference since $2w/\pi\xi \gg 1$ and it's inside a log, but it should still be explained.

*In Fig. 7, it would be good to clarify in the caption that B_0 is, in general, different for each data set. This doesn't become clear until one looks at Fig. 8, but the presentation in Fig. 7 makes it initially confusing.

*In Fig. 8, I don't understand the different values for the horizontal axis, ϕ/ϕ_s , for the data points. It looks like some of the samples have different lengths, but do some of them also have different widths? If not, ϕ_s should be the same for all the samples based on the definition in the main paper. Also, is the cooling field $B_{cd}=820$ uT for all of these points or is this being varied? Does Fig. 8 only contain values extracted from FC measurements? Otherwise, if ZFC measurements are combined with FC data here, this would be potentially confusing.

*Also in Fig. 8, I find it surprising that some of the B_0 values are larger than B_{cd} — is this consistent with the authors' VQ model? This is difficult to visualize how a VQ would arise in the intuitive picture of the sketches of the Gibbs free energy in Fig. 2(e).

*The measurements of coherence properties for the VQs away from the sweet spot in Fig. 14(a) are interesting and would be worth referring to explicitly in the main paper. Determining the dominant decoherence sources for these VQs and comparing these with the physics of decoherence for conventional superconducting qubits is an important problem.

*In Fig. 15(b), why are there no data points for $f_q(B-B_0)$ and only a fit line? It seems odd to only plot the fit without the data.

*In Fig. 2(e), the authors should clarify that $x = 0$ is at one edge of the resonator, not at the center (as in Ref. [31]).

*In Fig. 3(b), f^{beat} labeled on plot should match the text in the caption: 1.9 vs. 2 MHz.

Referee #3

(Remarks to the Author)

I have read the manuscript by Nambisan et al., which describes measurements on the quantum coherent properties of vortices placed in highly disordered superconductors. The manuscript is well written and the experimental results are clear. There is extensive supporting data and theoretical derivation in the appendices.

Coupling vortices to a resonant circuit is an exciting experimental feat, and it is impressive to see the frequency control as well as coherence properties being measured. The presented work will open up a new avenue for studying vortex physics.

The novel connection to a possible quantum information platform is not clear however. Vortices are known to be quantum objects, and have been well-documented to interact with film disorder, through their mobility, as well as with the superconducting electron system. This makes them fragile, for example by taking in excess quasiparticles [Ref 11 in the manuscript]. And while disordered films contain the pinning sites that are used here, such films are also found to have elevated quasiparticle numbers [Ref 50 in the manuscript]. The phase coherence being very low, even on the flux-insensitive point, is unfortunate, but perhaps not a real surprise. Further questions remain on how to perform individual control for entangling gates and flux tuning, which is not trivial as vortices are small objects.

The experimental results are highly interesting to the physics community and prompt follow-up research to investigate the relations with the disorder and electron system. As such, the manuscript is a strong candidate for more a physics-oriented journal.

Minor comments:

- Line 154 mentions AQRM, but it's not explicitly defined.

Version 1:

Reviewer comments:

Referee #1

(Remarks to the Author)

Let me start by stating that the results of this paper are significant and worthy of publication in Nature. It advances a new platform of vortices in superconductors, specifically granular aluminium as qubits.

The authors have addressed most of my concerns. The paper can nevertheless be strengthened along the following directions:

[1] It would be useful to have some estimates on the size of the aluminium grains and the distance between grains for which vortices behave as a coherent quantum object with a non-dissipative insulating core. Some discussion on when a Josephson vortex might crossover to an Abrikosov vortex may be useful to understand the design principles for vortex qubits.

[2] The authors could consider referring to Bouadim, et al Nature Phys. 7, 884-889 (2011), Ghosal et. al Phys. Rev. B 65, 014501 (2002); Phys. Rev. Lett. 81 3940 (1998) as the seminal references on emergent granularity.

[3] Small suggestions:

(a) On the first page "quantum non-demolishing readout" -- should this be "quantum non-demolition readout"?

(b) On page (2) first column "hyperbolic field dispersion eccentricity" should be explained. In general, the narrative in the main text should be more easily accessible to a broader readership and technical terms should be explained either in the main text or in the supplement as appropriate.

(c) For future studies, would it be useful to image the vortices and their cores using STM? if so, the authors could mention that.

Referee #2

(Remarks to the Author)

The revised manuscript of Nambisan et al. satisfactorily addresses the concerns I raised in my review of the previous version. The added text and figures help to clarify important aspects of the experiment and analysis. I remain enthusiastic about this work and recommend publication.

I only have one quite minor edit — on line 227, it would be better to use parentheses around the factor of 2π in front of the natural log to avoid confusion from the use of the solidus.

Britton Plourde

Quantum Coherence in Superconducting Vortex States

Ameya Nambisan and Ioan M. Pop

RESPONSE TO REFEREES

We want to start by thanking all three referees for their words of appreciation and their interest in the manuscript. We also welcome their constructive feedback, which we feel has significantly improved the clarity of the manuscript.

In the following, we provide a point-by-point response and outline the corresponding changes to the manuscript. The referees' comments are listed in blue, and our response is in black. The modifications to the manuscript are highlighted in the PDF.

REFEREE A

General comment

The paper contains important results that are especially important for the growing field of quantum coherence and manipulation. However, the paper is not well written for reasons I outline below. The authors would like to claim that the vortices in granular aluminum (grAl) behave like qubits, and not the standard Abrikosov vortices with dissipative metallic cores. This feature results in long coherence times and therefore enables various functionalities.

Reply: We thank the reviewer for recognizing the significance of the results and for the candid feedback on clarity and structure. We agree that the original manuscript did not communicate the central message with sufficient clarity for a wide audience, such as that of the journal Nature. In the revised manuscript, we substantially improved the organization and readability, which we detail below in the reviewer's specific comments.

Referee comment A.1 — To get this message across it is necessary to first describe the experimental results without any reference to theory.

Reply: We agree with the reviewer and thank them for this important remark. In the revised manuscript, we have restructured the presentation such that the experimental results are presented first in a fully self-contained manner, with minimal references to the theoretical models. Specifically, we implemented the following changes:

- 1) The discussion of the origin of the double-well potential in the Gibbs free energy, previously included in Fig. 2, has been moved to a new Fig. 4, with the corresponding discussion relocated to the end of the revised manuscript.
- 2) The theoretical discussion related to the introduction of Pearl vortices in grAl resonators has been shortened and merged with the discussion accompanying Fig. 4.
- 3) Fig. 1 has been reworked to focus exclusively on a comparison between zero-field-cooled and field-cooled experiments, with all references to theoretical predictions removed, thereby emphasizing the experimental observations alone.

Referee comment A.2 — It will be useful to have a comparison of various time scales (coherence, switching) in current platforms, a table will help. This will help position grAl as a new platform for certain functionalities.

Reply: We thank the referee for this suggestion. In the revised manuscript, we now explicitly benchmark the vortex qubit (VQ) against state-of-the-art superconducting qubit platforms, which constitute the most relevant point of comparison. In particular, we compare the measured relaxation and coherence times to those of superconducting flux qubits, thereby positioning grAl as a distinct platform with competitive lifetimes. Consistent with comment B11 from Reviewer B, we further include a comparison of dephasing away from the sweet spot to flux-noise-induced dephasing observed in superconducting flux qubits.

Specifically, we now state: "The measured VQ lifetime T_1 is competitive with superconducting flux qubits (Kjaergaard *et al.*, 2020; Yan *et al.*, 2016), whereas the coherence T_2^* , T_2^{echo} remains more modest, in line with flux qubit devices realized entirely from disordered superconductors (Purmessur *et al.*, 2025; Rieger *et al.*, 2023). Away from the sweet spot, both T_2^*

and T_2^{echo}) decrease (cf. App.J), consistent with flux-noise-limited dephasing in loop-based superconducting circuits and motivating a detailed comparison to established flux-noise mechanisms (Günzler *et al.*, 2025; Kjaergaard *et al.*, 2020; Yan *et al.*, 2016). ”

Referee comment A.3 — The main results of the paper should be described at the beginning before the experimental system and measurements are described. These should also be highlighted in the abstract.

Reply: We agree and also following a similar remark of the Editor we revised the manuscript to foreground the main findings. Specifically, we rewrote the abstract to explicitly state the key results and their significance to a broad audience. In addition, we reworked the “Here we show” paragraph at the end of the Introduction to provide a concise, results-first summary before introducing the experimental system and measurement details.

Referee comment A.4 — The role of theory is best described after the experiments. It is important to note that the predictions of vortices with insulating cores have been made by other groups as well in disordered superconductors and these should be cited. In these cases disorder creates emergent granularity that behaves very similarly to grAl.

Reply: We fully agree and thank the reviewer for bringing a broader literature context to our attention. In the revised manuscript, we have expanded our discussion of gapped vortex cores in granular superconductors by adding the following sentence:

“[...] a closely related regime has also been predicted for strongly disordered superconductors, where emergent superconducting islands (Trivedi *et al.*, 2012) host vortices with insulating cores (Datta *et al.*, 2023).”

Moreover, we included these references in the reworked abstract of the revised manuscript.

Referee comment A.5 — It will be useful to compare with disordered superconductors and the coherence time scales of Josephson vortices trapped with insulating cores.

Reply: We thank the reviewer for their suggestion. To the best of our knowledge, quantitative coherence metrics for Josephson vortices with insulating cores have not yet been reported in the literature, precluding a direct comparison. To address the broader point regarding disordered superconductors, we instead benchmark the vortex qubit against superconducting circuits realized entirely from disordered materials, including NbN constriction-based devices and granular aluminum (grAl) nanjunction circuits. This comparison is also aligned with Referee comment A.2

Accordingly, we now state in the revised manuscript: “The measured VQ lifetime T_1 is competitive with superconducting flux qubits (Kjaergaard *et al.*, 2020; Yan *et al.*, 2016), whereas the coherence T_2^* , T_2^{echo} remains more modest, in line with flux qubit devices realized entirely from disordered superconductors (Purmessur *et al.*, 2025; Rieger *et al.*, 2023).”

REFEREE B
General comment

The manuscript of Nambisan et al. describes intriguing measurements of granular Al (grAl) microwave resonators in magnetic fields. For sufficiently large magnetic fields, the authors observe a microwave response of the resonator that behaves in a quite similar way to a Josephson-junction based flux qubit coupled to a resonator. This includes coherent Rabi oscillations characteristic of a driven two-level system, as well as measurements of energy relaxation times, which are on par with good conventional superconducting qubits, and dephasing rates. However, the device measured here is elegantly simple, merely a narrow strip of grAl with no microfabricated Josephson junctions. The results are interpreted in terms of a model of vortices trapped in random pinning wells, with one vortex able to tunnel between neighboring wells while coupling to the zero-point currents in the microwave resonator. Unlike Abrikosov vortices in conventional superconductors, where dissipation in the vortex core disfavors such quantum coherent effects, a recent theory for vortices in grAl films, described in Ref. [1], predicts a mini-gap in the vortex core bound states and thus greatly reduced dissipation.

The results presented here are rather amazing. The possibility of quantum mechanical effects in the motion of a vortex in a superconductor have been debated for many years, with some prior evidence for incoherent vortex tunneling but nothing demonstrating any sort of coherence that I am aware of. The clear two-level behavior measured here demonstrates that something quite surprising is happening in the vortex state of this grAl resonator. Although the authors present a mostly plausible model for what is happening with the vortices to explain their measurements, I am not completely convinced by all of the details of the model. Nonetheless, the results are so striking, and, for the most part, carefully executed, and I believe will inspire further experimental work and significant theoretical investigations, potentially addressing long outstanding questions such as the effective mass of a vortex in a superconductor. Thus, I recommend publication following a series of edits that I describe below.

Reply: We thank the reviewer for their enthusiastic evaluation and in the following we address their specific questions.

Referee comment B.1 — My primary concern is with the interpretation in terms of the tunneling dynamics of a single vortex. The authors argue in Appendix I that intervortex interactions are relatively weak so that the motion of the vortices can be considered independently. Each time the resonator is cycled above T_c and cooled in magnetic field B_{cd} , there are of order 100 vortices trapped in the resonator. The picture presented is that most of the time just one of these 100 vortices exhibits quantum coherent tunneling dynamics that couples to the resonator. With each repeated magnetic field cycling above T_c in the same cooling field B_{cd} , the authors observe different parameters, such as tunneling energy, field dispersion, and sweet spot offset for the VQ (vortex qubit). The fact that there are different values for each field cycle seems reasonable since the vortex trapping locations will depend on microscopic details of the pinning configuration in these highly disordered films as well as details of the exact thermal trajectory for the field cycle, which I suspect is difficult to reproduce exactly. Thus, many of the 100 vortices will likely be trapped either in wells that are too deep or too far away from adjacent wells to allow for any tunneling.

What seems surprising is that the authors report primarily seeing one prominent VQ for each field cycle, with only occasional field cycles resulting in zero or multiple VQs. In this independent vortex picture, I would have expected many cycles with no VQs and some with multiple VQs, with single-VQ cycles being not especially favored. Can the authors be more quantitative? If there are enough field cycles of the same device, would it be possible to make a histogram of the number of VQs observed on each cycle?

It would also be useful to have in the Appendix some plots equivalent to Fig. 1(d) for an example cycle with 0 VQs and another with 2 (or more) VQs.

Reply: We thank the reviewer for their careful reading and accurate summary of the vortex-trapping picture, which we fully share. We agree that, in an independent-vortex scenario with ~ 100 trapped vortices per field-cooling cycle, the distribution of observable VQ signatures per cycle is an important quantitative check. To settle this crucial point, we decided to perform additional measurements. We performed a total of 100 additional cool-down & warm-up cycles, enabled by our fast turnaround dilution cryostat: 99 field-cooling cycles and one zero-field reference. We added the measured statistics on the number of observed VQ signatures per field-cooling cycle in Appendix C. The resulting histogram (new Fig. 8a) shows that, above a threshold, we consistently observe vortex-related signatures in the grAl resonator response. Importantly, this new analysis shows a more nuanced situation. While we obtain single-VQ states strongly coupled to the resonator (i.e. the

main text scenario) in 54% of cooldowns at $\phi/\phi_S = 0.9$, in the rest of the cool-downs the number of coupled VQs cannot be determined unambiguously. In these cases we observe either discrete frequency jumps in the resonator response, a smooth frequency shift relative to the zero-field-cooled reference, or multiple avoided crossings. These features clearly indicate vortex trapping but do not uniquely identify the number of coupled VQs. Resolving such cycles would require time-intensive two-tone spectroscopy, which is beyond the scope of the present work. In light of this, we removed the sentence “Repeated field coolings typically result in a single prominent avoided crossing, though we occasionally observe zero or multiple” from the revised manuscript. Finally, following the referee’s suggestion, we added representative magnetic-field sweeps (new Fig. 8b–f) that provide direct examples complementary to Fig. 1(d), including a cycle with no clear avoided crossing and cycles exhibiting multiple avoided level crossings.

Referee comment B.2 — The value of $B_{cd} = 820 \mu\text{T}$ is indeed what one would calculate from the standard expression for the threshold cooling field from Refs. [30,31] for $\xi = 7 \text{ nm}$, but was there an attempt to check this experimentally, such as repeated scans like Fig. 1(b,d) for different cooling fields? I appreciate that cycling above and below T_c in different cooling fields is time consuming and nontrivial on a dilution fridge, but it would be interesting to see if there are at least some cooling fields below $820 \mu\text{T}$ that don’t result in the trapping of any vortices. Also, I would expect that there should be a downwards frequency shift in the resonance when the cooling field is greater than the threshold field, even if there are no VQs, coming from the reactance of the trapped vortices.

Reply: We agree with the referee that an experimental verification of the vortex-entry threshold is valuable, even though repeated thermal cycling in a dilution refrigerator is technically demanding. To address this point and in line with our reply to comment B1, we performed 99 field-cooling cycles with cooling fields spanning the vicinity of the expected threshold, and we added the resulting statistics and representative field sweeps to the revised manuscript (new Fig. 8; Appendix C). These data show a clear qualitative change in the resonator response as the cooling field is increased. For $\phi/\phi_S = 0.75$, the field-dependent response is indistinguishable from the zero-field-cooled reference, consistent with the absence of trapped vortices. Accordingly, for all cycles with $\phi/\phi_S \geq 0.9$, we observe signatures of vortices trapped in the grAl resonator. These include avoided level crossings when a coupled VQ is present, and, as anticipated by the reviewer, a downward frequency shift relative to the zero-field-cooled reference even in cycles without a resolvable avoided crossing, consistent with the reactive contribution of trapped vortices. Based on these observations, we infer that the onset of vortex trapping occurs between $\phi/\phi_S = 0.75$ and $\phi/\phi_S = 0.9$.

Referee comment B.3 — I don’t fully understand the authors’ statement on line 100. It seems from their earlier arguments that cooling in $\phi = \phi_S$ should just result in a number of vortices per square of 1, not a range from 1-3. This should be clarified.

Reply: We thank the reviewer for pointing out this potential confusion. In the manuscript, $\phi = B_c w^2 / \Phi_0$ denotes the applied flux through a square of w^2 area in the units of Φ_0 , whereas ϕ_S is the threshold for pearl vortices to be thermodynamically stable inside the strip (where a minimum in Gibbs energy develops) (Bronson *et al.*, 2006; Kogan, 1994; Stan *et al.*, 2004). Importantly, reaching $\phi = \phi_S$ does not imply that exactly one vortex occupies each w^2 area. Rather, vortex entry in narrow strips is governed by edge barriers and strong vortex-vortex interactions, leading to non-uniform filling and a vortex density that generally differs from the naive expectation $N = BA/\Phi_0$. Both theory and experiment show that, above threshold, the number of vortices per square can be smaller or larger than unity depending on geometry and field range. This behavior has been directly observed in prior experiments on thin superconducting strips, including scanning SQUID measurements, which find that even above the entry threshold, the vortex density does not scale one-to-one with applied flux (Bai *et al.*, 2025; Kuit *et al.*, 2008). The range $1 \leq N_{\odot} < 3$ quoted in the original manuscript was therefore intended only as a rough order-of-magnitude estimate based on these works, not as a strict counting rule.

To avoid ambiguity, and since this estimate is not required to follow the main results, we have removed the sentence quantifying the “1–3 vortices per square” range and the associated inset in Fig. 1(c), and moved the broader discussion of vortex-entry thresholds and buckling to the end of the manuscript, consistent with Referee comment A.1.

Referee comment B.4 — In Fig.1 and related scans, do the authors ever observe hysteresis if the B-field sweep is retraced, or are the ranges of these scans always chosen to stay below the entry/exit field for vortices to jump in/out of the resonator? This is briefly mentioned in the caption, but it would be good to add detail if possible. The authors describe complementary ZFC measurements where they can inject vortices into an initially flux-free state, which makes sense. But it should also be possible to change the vortex density for an FC state by varying the B field over a sufficiently wide range.

Reply: We agree with the referee that, in principle, the vortex configuration of a FC state can be modified by sweeping B over a sufficiently wide range. To explicitly test this, we extended the magnetic-field span for the device of Fig. 1 at the end of the measurement campaign (prior to warmup). In these extended-range sweeps (figure provided below), we observe that the characteristic offset field B_0 shifts and additional avoided crossings appear. We attribute these changes to a modified vortex density and/or rearranged vortex configuration induced by the larger field excursion, as anticipated by the referee.

Accordingly, for all FC data shown in the main text and in the newly added field-cooling statistics (Appendix C, Fig. 8), the sweep ranges are intentionally chosen to remain well below the field scale where vortex rearrangements become apparent. Within these ranges, we do not observe hysteresis between consecutive sweeps taken in opposite directions: retracing the B-field produces the same resonator response and avoided-crossing positions within experimental resolution. We have clarified this point in Appendix C in the revised manuscript:

Note that for all measurements with VQ count = 0 or 1 shown in Fig.8, we observe no hysteresis between two consecutive sweeps taken in opposite directions.

Referee comment B.5 — I don't completely follow the authors' derivation for the VQ-resonator interaction in Appendix H. It appears that this indicates that the coupling will be maximized for the situation with the two pinning sites of the VQ oriented along the y direction, i.e., the length of the resonator, and zero for an orientation along x. My intuition would be that the tunneling between pinning sites along the resonator width direction, i.e., x direction, would give the strongest g , since zero point currents in the resonator would exert a Lorentz force on the vortex that would tilt the bias of one pinning well relative to the other. This is also how the authors have drawn the example pinning sites in Fig. 2(e). I realize there are many unknown quantities in the derivation in Appendix H and this is mainly intended to provide a plausible argument for the order of magnitude of g , but it would be good to clarify this point about the orientation of the pinning sites for a VQ.

Reply: We thank the reviewer for this insightful comment and for highlighting the contrast between the semiclassical derivation in the original App. H and a Lorentz-force-based intuition for the vortex-resonator coupling. In our current understanding, these two coupling mechanisms could coexist, but at this point it is difficult to say more, given that we don't have a clear picture of the VQ states.

As the referee correctly notes, the purpose of Appendix H is not to provide a definitive microscopic theory, but rather to establish a physically reasonable order-of-magnitude estimate for the observed coupling strength g . In response to this comment, we have therefore rewritten Appendix H to focus on the simplest and most intuitive coupling mechanism following Ref. (Bulaevskii *et al.*, 2011), i.e. the Lorentz-like interaction (in the case of grAl mediated by kinetic inductance, not geometric inductance) between the VQ and the resonator zero point fluctuations of the current.

In this revised formulation, the coupling requires the orientation of the pinning sites to be along the width of the resonator (the same as in Appendix G), and yields $\tilde{g}/\omega_r \sim 1\%$, comparable to the experimentally extracted values.

Referee comment B.6 — In the appendix, it would be helpful to have a schematic of the dilution fridge setup and wiring. Was any magnetic shielding included to screen external magnetic fields so that zero current in the Helmholtz pair is close to zero total field? Was any IR shielding used around the waveguide containing the grAl resonator? This seems potentially challenging with the Helmholtz pair at 4 K, but perhaps could be related to the elevated effective temperatures for the VQs.

Reply: We thank the reviewer for this suggestion. We have now modified Fig. 5 to include the readout chain inside the cryostat and have edited the caption to include:

[Caption] The input line (marked 'in') is attenuated by 75 dB (including the attenuation of the lines) distributed at the different temperature stages, while the output ('out') is routed through a double-junction isolator and a high electron mobility transistor (HEMT) amplifier.

No additional magnetic shielding (e.g., μ -metal or superconducting shields) was employed in our setup. Nevertheless, under zero-field-cooled conditions, the resonator response is reproducible and provides a clear reference to identify the absolute zero-field point.

We also confirm that no dedicated infrared (IR) shielding was installed around the waveguide/sample enclosure. We agree that this likely contributes to the elevated effective VQ temperature 74 mK, consistent with similar observations in Refs. (Günzler *et al.*, 2025). This is now stated explicitly in the revised manuscript.

[APP. A] We attribute the elevated VQ temperature of $T_{\text{eff}} = 74$ mK (cf. 2(a)) to thermal photons leaking into the sample holder through the copper dowel due to the absence of further infrared shielding, consistent with elevated temperatures observed in a grAl fluxonium qubit measured in the same setup (Günzler *et al.*, 2025). We did not employ additional magnetic shielding; however, the zero-field-cooled resonator measurement (Fig 1(b)) provides an in-situ estimate of the offset magnetic field. Specifically, the maximum of the parabolic suppression of the resonator frequency indicates an offset field close to zero.

Improving IR shielding is a natural next step; however, as the reviewer notes, implementing an effective IR-tight enclosure is technically constrained by the proximity of the 4 K Helmholtz pair and the available space at that stage.

Referee comment B.7 — In Eq. (3), and also Eq. (G2), where does the +1 in the argument of the log term in Eq. (3) come from? This is not in Ref. [31], although it doesn't make much of a difference since $2w/\pi\xi \gg 1$ and it's inside a log, but it should still be explained.

Reply: We thank the reviewer for bringing this to our attention. The '+1' term is introduced as a regularization to prevent unphysical divergence as $x \rightarrow 0$ or $x \rightarrow w$. This is because the underlying London thin-film is not valid within distances of ξ close to the film edge (Kogan, 1994). In the revised manuscript, we therefore add Ref. (Bai *et al.*, 2025), where the same regularization is justified in the appendix.

Referee comment B.8 — In Fig. 7, it would be good to clarify in the caption that B_0 is, in general, different for each data set. This doesn't become clear until one looks at Fig. 8, but the presentation in Fig. 7 makes it initially confusing.

Reply: We agree with the reviewer and have revised the figure caption accordingly to clarify this point. The caption now reads: [Caption] Extracted VQ frequency with fits to the Hamiltonian (Eq. 2) as a function of magnetic field detuning from their respective sweet spots B_0 , which differ between preparation cycles (cf. Fig. 10b).

Referee comment B.9 — In Fig. 8, I don't understand the different values for the horizontal axis, ϕ/ϕ_s , for the data points. It looks like some of the samples have different lengths, but do some of them also have different widths? If not, ϕ_s should be the same for all the samples based on the definition in the main paper.

Also, is the cooling field $B_{\text{cd}} = 820$ μT for all of these points or is this being varied?

Does Fig. 8 only contain values extracted from FC measurements? Otherwise, if ZFC measurements are combined with FC data here, this would be potentially confusing.

Reply: We thank the referee for raising this point. All data shown in Fig. 8 (now Fig. 10) are obtained exclusively from field-cooling (FC) measurements; no ZFC data are combined in this figure. The horizontal axis ϕ/ϕ_s varies for two reasons. First, we intentionally varied the cooldown field B_{cd} between 780–1080 μT , (i.e., in the vicinity of the threshold) so ϕ/ϕ_s differs between data points. Second, while all resonators were nominally 3 μm wide, AFM measurements reveal small post-fabrication width-variations between wafers: samples A, B (wafer 1) have slanted edges with an average width of 3.15 μm , whereas samples C-F (wafer 2) have sharper edges with a measured width of 2.8 μm . Accordingly, Fig. 10 compiles 32 VQ spectra from six samples (A–F) across two wafers; the values of ϕ/ϕ_s reflect both the chosen cooldown field and the measured sample width. We clarified these points in Appendix C and in the revised caption, and we explicitly note in the text that B_{cd} is varied across the dataset.

The reworked part of the figure caption now reads:

[Caption] (a) Sweet-spot qubit frequency f_q , (b) offset field B_0 , (c) gyromagnetic ratio γ , and (d) VQ-resonator coupling g as a function of the field bias ϕ during field cooling, normalized to ϕ_s (cf. main text and Ref. (Kogan, 1994)). The values ϕ/ϕ_s are given by the cooldown field and each sample's respective width.

We also added one sentence into the discussion on Fig. 10 in App.C:

Multiple field-cooling cycles were performed with B_{cd} varied in the vicinity of the threshold field.

In App. C, we added a paragraph detailing the different samples A-F:

We measured VQs in six grAl resonators (samples A - F) fabricated on two wafers. Sample A is discussed in the main

text. Samples C - F were deposited in a different e-beam evaporator than samples A and B, yielding a sheet resistance of $R_{\square} = 1.2 \text{ k}\Omega/\square$ for a film-thickness of 30 nm. Although all resonators were nominally $3 \mu\text{m}$ wide, AFM measurements revealed post-fabrication variations: resonators A and B had slanted edges with an average width of $3.15 \mu\text{m}$ (cf. Fig. 7), whereas resonators C - F featured sharp edges and a measured width of $2.8 \mu\text{m}$.

Referee comment B.10 — Also in Fig. 8, I find it surprising that some of the B_0 values are larger than B_{cd} — is this consistent with the authors' VQ model? This is difficult to visualize how a VQ would arise in the intuitive picture of the sketches of the Gibbs free energy in Fig. 2(e).

Reply: We thank the referee for highlighting this potential confusion. In our model, B_0 denotes the degeneracy (sweet-spot) field at which the two localized vortex configurations (i.e., left/right pinning sites) have equal energy. Importantly, B_0 is not the vortex-entry field and is therefore not constrained to be smaller than the cooldown field B_{cd} (or the threshold field corresponding to ϕ_S). Rather, B_0 is determined by the balance between (i) the intrinsic asymmetry of the two pinning sites (different pinning depths at $B = 0$ and (ii) the field-dependent “background” contribution to the Gibbs free energy $G_1(B)$, which can tilt the relative energies of the two configurations. If one site is substantially deeper than the other at zero field, reaching degeneracy requires a large field-induced shift, which can occur at fields either below or above B_{cd} . We agree that this was not well visualized in the original schematic. In the revised manuscript, we therefore modified Fig. 2(e) (now Fig. 4) such that the background term $G_1(B)$ exhibits a minimum at $B = B_0$ (and does not coincide with a flat $G_1(B)$), making it explicit how degeneracy can occur.

We also clarified this directly in the caption:

[Caption] At the sweet spot (B_0 , which can be higher or lower than B_S ; cf. App.C), the double-well is degenerate, with VQ states forming symmetric and antisymmetric combinations of the localized wavefunctions, yielding $\hbar\omega_q = 2\Delta$.

Referee comment B.11 — The measurements of coherence properties for the VQs away from the sweet spot in Fig. 14(a) are interesting and would be worth referring to explicitly in the main paper. Determining the dominant decoherence sources for these VQs and comparing these with the physics of decoherence for conventional superconducting qubits is an important problem.

Reply: We agree with the reviewer that identifying the dominant dephasing mechanisms of VQs (particularly away from the sweet spot), and benchmarking them against decoherence in conventional superconducting qubits is an important direction. In loop-based superconducting circuits, flux-noise-limited dephasing is widely associated with fluctuating surface spins on the superconducting loop (Braumüller *et al.*, 2020; Günzler *et al.*, 2025; Kjaergaard *et al.*, 2020; Stern *et al.*, 2014; Yan *et al.*, 2016). Our observation that the VQ coherence degrades away from the sweet spot is qualitatively consistent with such flux-noise-limited dephasing, motivating a direct, quantitative comparison of noise mechanisms in future work.

In response, we now explicitly refer to these off-sweet-spot coherence data in the main text and highlight their connection to established flux-noise physics. Specifically, we added the following sentence:

Away from the sweet spot, both T_2^* and T_2^{echo} decrease (cf. App. J), consistent with flux-noise-limited dephasing in loop-based superconducting circuits and motivating a detailed comparison to established flux-noise mechanisms (Günzler *et al.*, 2025; Kjaergaard *et al.*, 2020; Yan *et al.*, 2016).

Referee comment B.12 — In Fig. 15(b), why are there no data points for $f_q(B - B_0)$ and only a fit line? It seems odd to only plot the fit without the data.

Reply: Following the referee's comment, Fig. 15(b) (now Fig. 16(b)) has been modified to include this information.

Referee comment B.13 — In Fig. 2(e), the authors should clarify that $x = 0$ is at one edge of the resonator, not at the center (as in Ref. [31]).

Reply: We thank the reviewer for pointing this out. In the revised manuscript, we updated Fig. 2(e) (now Fig. 4) to make the coordinate convention explicit: the vortex position x is measured from the resonator edge, such that $x = 0$ is at the edge (rather than at the center, as in Ref. (Bronson *et al.*, 2006)). To avoid ambiguity, we also indicate the resonator width with a single-headed arrow referenced to the edge. We additionally state this directly in the revised caption:

[Caption] The vortex position is measured from the edge, as indicated by the coordinate axis.

Referee comment B.14 — In Fig. 3(b), f^{beat} labeled on plot should match the text in the caption: 1.9 vs. 2 MHz.

Reply: In line with the reviewer's comment, we have corrected this inconsistency in Fig. 3(b) so that the value of f^{beat} matches the plot label and the caption (now both state 1.9 MHz).

REFeree C

I have read the manuscript by Nambisan et al., which describes measurements on the quantum coherent properties of vortices placed in highly disordered superconductors. The manuscript is well written and the experimental results are clear. There is extensive supporting data and theoretical derivation in the appendices. Coupling vortices to a resonant circuit is an exciting experimental feat, and it is impressive to see the frequency control as well as coherence properties being measured. The presented work will open up a new avenue for studying vortex physics.

Reply: We thank the referee for their careful reading of the manuscript and for their positive evaluation.

Referee comment C.1 — The novel connection to a possible quantum information platform is not clear however. Vortices are known to be quantum objects, and have been well-documented to interact with film disorder, through their mobility, as well as with the superconducting electron system. This makes them fragile, for example by taking in excess quasiparticles [Ref 11 in the manuscript]. And while disordered films contain the pinning sites that are used here, such films are also found to have elevated quasiparticle numbers [Ref 50 in the manuscript]. The phase coherence being very low, even on the flux-insensitive point, is unfortunate, but perhaps not a real surprise. Further questions remain on how to perform individual control for entangling gates and flux tuning, which is not trivial as vortices are small objects. The experimental results are highly interesting to the physics community and prompt follow-up research to investigate the relations with the disorder and electron system. As such, the manuscript is a strong candidate for more a physics-oriented journal.

Reply: We agree with the reviewer that, despite the exceptionally long T_1 values, establishing vortex states as a quantum computing platform remains a long and uncertain journey, though likely a scientifically exciting one. This is only the beginning: the first observation of quantum coherence in vortex states. Beyond a new quantum information platform, these results open up several research and technology avenues: in hybrid architectures, in magnetic sensing (similar to SQUID-on-cantilever magnetometers), and in material characterization (in-situ probe). Last, but not least, the measurement of quantum coherence in vortex states introduces a new dimension for the understanding of the superconductor-to-insulator transition in the Bosonic scenario.

In view of these arguments, and the fact that we rephrased the abstract and the introduction to make it more accessible for a wider audience, we believe our manuscript is a good fit for Nature.

Referee comment C.2 — Line 154 mentions AQRM, but it's not explicitly defined.

Reply: We thank the reviewer for pointing this out. The asymmetric quantum Rabi model (AQRM) has now been explicitly defined when it is first introduced in the main text.

REFERENCES

- Bai, Ruiheng, Aliakbar Sepehri, Yen-Lee Loh, Anne-Marie Valente-Feliciano, Anna Herr, Quentin Herr, and Katja C. Nowack (2025), “Flux trapping in NbTiN strips and structures,” arXiv preprint [arXiv:2503.14457](https://arxiv.org/abs/2503.14457).
- Braumüller, Jochen, Leon Ding, Antti P. Vepsäläinen, Youngkyu Sung, Morten Kjaergaard, Tim Menke, Roni Winik, David Kim, Bethany M. Niedzielski, Alexander Melville, Jonilyn L. Yoder, Cyrus F. Hirjibehedin, Terry P. Orlando, Simon Gustavsson, and William D. Oliver (2020), “Characterizing and Optimizing Qubit Coherence Based on SQUID Geometry,” *Phys. Rev. Appl.* **13** (5), 054079.
- Bronson, Eric, Martin P. Gelfand, and Stuart B. Field (2006), “Equilibrium configurations of pearl vortices in narrow strips,” *Physical Review B* **73** (14), 144501.
- Bulaevskii, L N, M. J. Graf, C. D. Batista, and V. G. Kogan (2011), “Vortex-induced dissipation in narrow current-biased thin-film superconducting strips,” *Phys. Rev. B* **83** (14), 144526.
- Datta, Anushree, Anurag Banerjee, Nandini Trivedi, and Amit Ghosal (2023), “Insulating vortex cores in disordered superconductors,” *Phys. Rev. B* **107** (14), L140502.
- Günzler, S, J. Beck, D. Rieger, N. Gosling, N. Zapata, M. Field, S. Geisert, A. Bacher, J. K. Hohmann, M. Spiecker, W. Wernsdorfer, and I. M. Pop (2025), “Spin environment of a superconducting qubit in high magnetic fields,” *Nat. Commun.* **16** (9564), 1–7.
- Kjaergaard, Morten, Mollie E. Schwartz, Jochen Braumüller, Philip Krantz, Joel I.-J. Wang, Simon Gustavsson, and William D. Oliver (2020), “Superconducting qubits: Current state of play,” *Annual Review of Condensed Matter Physics* **11** (Volume 11, 2020), 369–395.
- Kogan, Vladimir G (1994), “Pearl’s vortex near the film edge,” *Physical Review B* **49** (22), 15874–15878.

- Kuit, K H, J. R. Kirtley, W. Van Der Veur, C. G. Molenaar, F. J. G. Roesthuis, A. G. P. Troeman, J. R. Clem, H. Hilgenkamp, H. Rogalla, and J. Flokstra (2008), “Vortex trapping and expulsion in thin-film $\text{YBa}_2\text{Cu}_3\text{O}_{7-\delta}$ strips,” *Physical Review B* **77** (13), 134504.
- Purmessur, Cheeranjeev, Kaicheung Chow, Bernard van Heck, and Angela Kou (2025), “Operation of a high-frequency, phase-slip qubit,” *Nat. Commun.* [10.1038/s41467-025-66348-w](https://doi.org/10.1038/s41467-025-66348-w).
- Rieger, D, S. Günzler, M. Spiecker, P. Paluch, P. Winkel, L. Hahn, J. K. Hohmann, A. Bacher, W. Wernsdorfer, and I. M. Pop (2023), “Granular aluminium nanojunction fluxonium qubit,” *Nature Materials* **22**, 194–199.
- Stan, Gheorghe, Stuart B. Field, and John M. Martinis (2004), “Critical field for complete vortex expulsion from narrow superconducting strips,” *Physical Review Letters* **92** (9), 097003.
- Stern, M, G. Catelani, Y. Kubo, C. Grezes, A. Bienfait, D. Vion, D. Esteve, and P. Bertet (2014), “Flux qubits with long coherence times for hybrid quantum circuits,” *Physical Review Letters* **113** (12), 123601.
- Trivedi, Nandini, Yen Lee Loh, Karim Bouadim, and Mohit Randeria (2012), “Emergent granularity and pseudogap near the superconductor-insulator transition,” *J. Phys. Conf. Ser.* **376** (1), 012001.
- Yan, Fei, Simon Gustavsson, Archana Kamal, Jeffrey Birenbaum, Adam P. Sears, David Hover, Ted J. Gudmundsen, Danna Rosenberg, Gabriel Samach, S. Weber, Jonilyn L. Yoder, Terry P. Orlando, John Clarke, Andrew J. Kerman, and William D. Oliver (2016), “The flux qubit revisited to enhance coherence and reproducibility,” *Nature Communications* **7** (1), 12964.

Quantum Coherence in Superconducting Vortex States

Ameya Nambisan and Ioan M. Pop

RESPONSE TO REFEREES

We thank all reviewers for their positive feedback and their recommendation to publish. In the following we offer a point-by-point reply to their remaining comments and we describe the corresponding changes to the manuscript. The reviewers' comments are listed in blue and our response in black.

REFEREE A

General comment

Let me start by stating that the results of this paper are significant and worthy of publication in Nature. It advances a new platform of vortices in superconductors, specifically granular aluminium as qubits.

Reply: We thank the reviewer for their positive assessment and for emphasizing the significance of our results.

Referee comment A.1 — It would be useful to have some estimates on the size of the aluminium grains and the distance between grains for which vortices behave as a coherent quantum object with a non-dissipative insulating core. Some discussion on when a Josephson vortex might crossover to an Abrikosov vortex may be useful to understand the design principles for vortex qubits.

Reply: We agree with the reviewer that identifying which microstructural parameters enable a vortex to behave as a coherent, weakly dissipative ("gapful-core") quantum object is an important open question and directly relevant for vortex-qubit design. A concrete theoretical criterion has been predicted for granular superconductors (Kiselov *et al.*, 2023): the quasiparticle spectrum in the vortex core becomes gapful when the superconducting coherence length ξ , becomes comparable to the intergrain spacing l (and a growing minigap for $\xi \lesssim l$). However, a definitive answer would require a systematic study of different resistivity grAl films in which the microscopic grain size and intergrain spacing vary, and is a promising direction for future research.

Following the reviewer's comment, we added to the conclusion the following sentence: "This supports a picture of grAl as a 3D network of Josephson junctions, expected to host gapful-core vortices once the coherence length ξ becomes comparable to the intergrain spacing l , with a growing minigap for $\xi \lesssim l$."

Referee comment A.2 — The authors could consider referring to Bouadim, et al Nature Phys. 7, 884-889 (2011), Ghosal et. al Phys. Rev. B 65, 014501 (2002); Phys. Rev. Lett. 81 3940 (1998) as the seminal references on emergent granularity.

Reply: We thank the referee for drawing our attention to these seminal works on emergent granularity. Restricted by Nature's 50-reference limit, we cannot add all five citations. We have therefore added Bouadim et al. (Nat. Phys. 7, 884–889 (2011)) and, to accommodate this, replaced the conference-proceedings citation (J. Phys. Conf. Ser. 376, 012001 (2012))

Referee comment A.3 — On the first page "quantum non-demolishing readout" – should this be "quantum non-demolition readout"?

Reply: We thank the reviewer for pointing this out and have corrected it accordingly; the text now reads 'quantum non-demolition readout' on the first page.

Referee comment A.4 — On page (2) first column "hyperbolic field dispersion eccentricity" should be explained. In general, the narrative in the main text should be more easily accessible to a broader readership and technical terms should be explained either in the main text or in the supplement as appropriate.

Reply: We thank the referee for pointing this out. To improve accessibility, we replaced the term “hyperbolic field dispersion eccentricity” with “slope of the hyperbolic field dispersion” in the main text.

Referee comment A.5 — For future studies, would it be useful to image the vortices and their cores using STM? if so, the authors could mention that.

Reply: We agree that STM imaging of vortices and their cores would be valuable in future work, and we already note this directly before the conclusion paragraph of the manuscript.

REFEREE B

General comment

The revised manuscript of Nambisan et al. satisfactorily addresses the concerns I raised in my review of the previous version. The added text and figures help to clarify important aspects of the experiment and analysis. I remain enthusiastic about this work and recommend publication.

Reply: Thank you for your appreciation and your interest in our work.

Referee comment B.1 — I only have one quite minor edit — on line 227, it would be better to use parentheses around the factor of $2/\pi$ in front of the natural log to avoid confusion from the use of the solidus.

Reply: We agree with the reviewer and have added parentheses around the factor $(2/\pi)$ in the revised manuscript to avoid any ambiguity.

REFERENCES

Kiselov, Dmitry, Mikhail A. Skvortsov, and Mikhail V. Feigel'man (2023), “Gapful electrons in a vortex core in granular superconductors,” *SciPost Physics* **15** (1), 10.21468/scipostphys.15.1.008.